# Learning General Purpose Distributed Sentence Representations via Large Scale Multi-task Learning

**Sandeep Subramanian**[1,2,3,*] **Adam Trischler**[3]**, Yoshua Bengio**[1,2,4] **& Christopher J Pal**[1,5]
[1] Montréal Institute for Learning Algorithms (MILA)
[2] Université de Montréal
[3] Microsoft Research Montreal
[4] CIFAR Senior Fellow
[5] École Polytechnique de Montréal

## Abstract

A lot of the recent success in natural language processing (NLP) has been driven by distributed vector representations of words trained on large amounts of text in an unsupervised manner. These representations are typically used as general purpose features for words across a range of NLP problems. However, extending this success to learning representations of sequences of words, such as sentences, remains an open problem. Recent work has explored unsupervised as well as supervised learning techniques with different training objectives to learn general purpose fixed-length sentence representations. In this work, we present a simple, effective multi-task learning framework for sentence representations that combines the inductive biases of diverse training objectives in a single model. We train this model on several data sources with multiple training objectives on over 100 million sentences. Extensive experiments demonstrate that sharing a single recurrent sentence encoder across weakly related tasks leads to consistent improvements over previous methods. We present substantial improvements in the context of transfer learning and low-resource settings using our learned general-purpose representations.

## 1 Introduction

Transfer learning has driven a number of recent successes in computer vision and NLP. Computer vision tasks like image captioning (Xu et al., 2015) and visual question answering typically use CNNs pretrained on ImageNet (Krizhevsky et al., 2012; Simonyan & Zisserman, 2014) to extract representations of the image, while several natural language tasks such as reading comprehension and sequence labeling (Lample et al., 2016) have benefited from pretrained word embeddings (Mikolov et al., 2013; Pennington et al., 2014) that are either fine-tuned for a specific task or held fixed.

Many neural NLP systems are initialized with pretrained word embeddings but learn their representations of *words in context* from scratch, in a task-specific manner from supervised learning signals. However, learning these representations reliably from scratch is not always feasible, especially in low-resource settings, where we believe that using general purpose sentence representations will be beneficial.

Some recent work has addressed this by learning general-purpose sentence representations (Kiros et al., 2015; Wieting et al., 2015; Hill et al., 2016; Conneau et al., 2017; McCann et al., 2017; Jernite et al., 2017; Nie et al., 2017; Pagliardini et al., 2017). However, there exists no clear consensus yet on what training objective or methodology is best suited to this goal.

Understanding the inductive biases of distinct neural models is important for guiding progress in representation learning. Shi et al. (2016) and Belinkov et al. (2017) demonstrate that neural machine translation (NMT) systems appear to capture morphology and some syntactic properties. Shi

---

[*]Work done while author was an intern at Microsoft Research Montreal

et al. (2016) also present evidence that sequence-to-sequence parsers (Vinyals et al., 2015) more strongly encode source language syntax. Similarly, Adi et al. (2016) probe representations extracted by sequence autoencoders, word embedding averages, and skip-thought vectors with a multi-layer perceptron (MLP) classifier to study whether sentence characteristics such as length, word content and word order are encoded.

To generalize across a diverse set of tasks, it is important to build representations that encode several aspects of a sentence. Neural approaches to tasks such as skip-thoughts, machine translation, natural language inference, and constituency parsing likely have different inductive biases. Our work exploits this in the context of a simple one-to-many multi-task learning (MTL) framework, wherein a single recurrent sentence encoder is shared across multiple tasks. We hypothesize that sentence representations learned by training on a reasonably large number of weakly related tasks will generalize better to novel tasks unseen during training, since this process encodes the inductive biases of multiple models. This hypothesis is based on the theoretical work of Baxter et al. (2000). While our work aims at learning *fixed-length* distributed sentence representations, it is not always practical to assume that the entire "meaning" of a sentence can be encoded into a fixed-length vector. We merely hope to capture some of its characteristics that could be of use in a variety of tasks.

The primary contribution of our work is to combine the benefits of diverse sentence-representation learning objectives into a single multi-task framework. To the best of our knowledge, this is the first large-scale reusable sentence representation model obtained by combining a set of training objectives with the level of diversity explored here, i.e. multi-lingual NMT, natural language inference, constituency parsing and skip-thought vectors. We demonstrate through extensive experimentation that representations learned in this way lead to improved performance across a diverse set of novel tasks not used in the learning of our representations. Such representations facilitate low-resource learning as exhibited by significant improvements to model performance for new tasks in the low labelled data regime - achieving comparable performance to a few models trained from scratch using only 6% of the available training set on the Quora duplicate question dataset.

## 2 RELATED WORK

The problem of learning distributed representations of phrases and sentences dates back over a decade. For example, Mitchell & Lapata (2008) present an additive and multiplicative linear composition function of the distributed representations of individual words. Clark & Pulman (2007) combine symbolic and distributed representations of words using tensor products. Advances in learning better distributed representations of words (Mikolov et al., 2013; Pennington et al., 2014) combined with deep learning have made it possible to learn complex non-linear composition functions of an arbitrary number of word embeddings using convolutional or recurrent neural networks (RNNs). A network's representation of the last element in a sequence, which is a non-linear composition of all inputs, is typically assumed to contain a squashed "summary" of the sentence. Most work in supervised learning for NLP builds task-specific representations of sentences rather than general-purpose ones.

Notably, skip-thought vectors (Kiros et al., 2015), an extension of the skip-gram model for word embeddings (Mikolov et al., 2013) to sentences, learn re-usable sentence representations from weakly labeled data. Unfortunately, these models take weeks or often months to train. Hill et al. (2016) address this by considering faster alternatives such as sequential denoising autoencoders and shallow log-linear models. Arora et al. (2016), however, demonstrate that simple word embedding averages are comparable to more complicated models like skip-thoughts. More recently, Conneau et al. (2017) show that a completely supervised approach to learning sentence representations from natural language inference data outperforms all previous approaches on transfer learning benchmarks. Here we use the terms "transfer learning performance" on "transfer tasks" to mean the performance of sentence representations evaluated on tasks unseen during training. McCann et al. (2017) demonstrated that representations learned by state-of-the-art large-scale NMT systems also generalize well to other tasks. However, their use of an attention mechanism prevents the learning of a single fixed-length vector representation of a sentence. As a result, they present a bi-attentive classification network that composes information present in all of the model's hidden states to achieve improvements over a corresponding model trained from scratch. Jernite et al. (2017) and Nie et al. (2017)

demonstrate that discourse-based objectives can also be leveraged to learn good sentence representations.

Our work is most similar to that of Luong et al. (2015), who train a many-to-many sequence-to-sequence model on a diverse set of weakly related tasks that includes machine translation, constituency parsing, image captioning, sequence autoencoding, and intra-sentence skip-thoughts. There are two key differences between that work and our own. First, like McCann et al. (2017), their use of an attention mechanism prevents learning a fixed-length vector representation for a sentence. Second, their work aims for improvements on the same tasks on which the model is trained, as opposed to learning re-usable sentence representations that transfer elsewhere.

We further present a fine-grained analysis of how different tasks contribute to the encoding of different information signals in our representations following work by Shi et al. (2016) and Adi et al. (2016).

Hashimoto et al. (2016) similarly present a multi-task framework for textual entailment with task supervision at different levels of learning. "Universal" multi-task models have also been successfully explored in the context of computer vision problems (Kokkinos, 2016; Eigen & Fergus, 2015).

## 3 SEQUENCE-TO-SEQUENCE LEARNING

Five out of the six tasks that we consider for multi-task learning are formulated as sequence-to-sequence problems (Cho et al., 2014; Sutskever et al., 2014). Briefly, sequence-to-sequence models are a specific case of encoder-decoder models where the inputs and outputs are sequential. They directly model the conditional distribution of outputs given inputs $P(\mathbf{y}|\mathbf{x})$. The input $\mathbf{x}$ and output $\mathbf{y}$ are sequences $x_1, x_2, \ldots, x_m$ and $y_1, y_2 \ldots, y_n$. The encoder produces a fixed length vector representation $\mathbf{h_x}$ of the input, which the decoder then conditions on to generate an output. The decoder is auto-regressive and breaks down the joint probability of outputs into a product of conditional probabilities via the chain rule:

$$P(\mathbf{y}|\mathbf{x}) = \prod_{i=1}^{n} P(y_i|y_{<i}, \mathbf{h_x})$$

Cho et al. (2014) and Sutskever et al. (2014) use encoders and decoders parameterized as RNN variants such as Long Short-term Memory (LSTMs) (Hochreiter & Schmidhuber, 1997) or Gated Recurrent Units (GRUs) (Chung et al., 2014). The hidden representation $\mathbf{h_x}$ is typically the last hidden state of the encoder RNN.

Bahdanau et al. (2014) alleviate the gradient bottleneck between the encoder and the decoder by introducing an attention mechanism that allows the decoder to condition on every hidden state of the encoder RNN instead of only the last one. In this work, as in Kiros et al. (2015); Hill et al. (2016), we do not employ an attention mechanism. This enables us to obtain a single, fixed-length, distributed sentence representation. To diminish the effects of vanishing gradient, we condition every decoding step on the encoder hidden representation $\mathbf{h_x}$. We use a GRU for the encoder and decoder in the interest of computational speed. The encoder is a bidirectional GRU while the decoder is a unidirectional conditional GRU whose parameterization is as follows:

$$\mathbf{r_t} = \sigma(\mathbf{W_r}x_t + \mathbf{U_r h_{t-1}} + \mathbf{C_r h_x}) \tag{1}$$

$$\mathbf{z_t} = \sigma(\mathbf{W_z}x_t + \mathbf{U_z h_{t-1}} + \mathbf{C_z h_x}) \tag{2}$$

$$\tilde{\mathbf{h_t}} = \tanh(\mathbf{W_d}x_t + \mathbf{U_d}(\mathbf{r_t} \odot \mathbf{h_{t-1}}) + \mathbf{C_d h_x}) \tag{3}$$

$$\mathbf{h_{t+1}} = (1 - \mathbf{z_t}) \odot \mathbf{h_{t-1}} + \mathbf{z_t} \odot \tilde{\mathbf{h_t}} \tag{4}$$

The encoder representation $\mathbf{h_x}$ is provided as conditioning information to the reset gate, update gate and hidden state computation in the GRU via the parameters $\mathbf{C_r}$, $\mathbf{C_z}$ and $\mathbf{C_d}$ to avoid attenuation of information from the encoder.

### 3.1 MULTI-TASK SEQUENCE-TO-SEQUENCE LEARNING

Dong et al. (2015) present a simple one-to-many multi-task sequence-to-sequence learning model for NMT that uses a shared encoder for English and task-specific decoders for multiple target languages.

Luong et al. (2015) extend this by also considering many-to-one (many encoders, one decoder) and many-to-many architectures. In this work, we consider a one-to-many model since it lends itself naturally to the idea of combining inductive biases from different training objectives. The same bidirectional GRU encodes the input sentences from different tasks into a compressed summary $\mathbf{h_x}$ which is then used to condition a task-specific GRU to produce the output sentence.

# 4    TRAINING OBJECTIVES & EVALUATION

Our motivation for multi-task training stems from theoretical insights presented in Baxter et al. (2000). We refer readers to that work for a detailed discussion of results, but the conclusions most relevant to this discussion are (i) that learning multiple related tasks jointly results in good generalization as measured by the number of training examples required per task; and (ii) that inductive biases learned on sufficiently many training tasks are likely to be good for learning novel tasks drawn from the same environment.

We select the following training objectives to learn general-purpose sentence embeddings. Our desiderata for the task collection were: sufficient diversity, existence of fairly large datasets for training, and success as standalone training objectives for sentence representations.

**Skip-thought vectors**    Skip-thought vectors (Kiros et al., 2015) are an extension of skip-gram word embedding models to sentences. The task typically requires a corpus of contiguous sentences, for which we use the BookCorpus (Zhu et al., 2015). The learning objective is to simultaneously predict the next and previous sentences from the current sentence. The encoder for the current sentence and the decoders for the previous (STP) and next sentence (STN) are typically parameterized as separate RNNs. We also consider training skip-thoughts by predicting only the next sentence given the current, motivated by results in Tang et al. (2017) where it is demonstrated that predicting the next sentence alone leads to comparable performance.

**Neural Machine Translation**    Sutskever et al. (2014) and Cho et al. (2014) demonstrated that NMT can be formulated as a sequence-to-sequence learning problem where the input is a sentence in the source language and the output is its corresponding translation in the target language. We use a parallel corpus of around 4.5 million English-German (De) sentence pairs from WMT15 and 40 million English-French (Fr) sentence pairs from WMT14. We train our representations using multiple target languages motivated by improvements demonstrated by Dong et al. (2015).

**Constituency Parsing (linearized parse tree construction)**    Vinyals et al. (2015) demonstrated that a sequence-to-sequence approach to constituency parsing is viable. The input to the encoder is the sentence itself and the decoder produces its linearized parse tree. We train on 3 million weakly labeled parses obtained by parsing a random subset of the 1-billion word corpus with the Puck GPU parser[1] along with gold parses from sections 0-21 of the WSJ section of Penn Treebank. Gold parses are duplicated 5 times and shuffled in with the weakly labeled parses to have a roughly 1:5 ratio of gold to noisy parses.

**Natural Language Inference**    Natural language inference (NLI) is a 3-way classification problem. Given a premise and a hypothesis sentence, the objective is to classify their relationship as either entailment, contradiction, or neutral. In contrast to the previous tasks, we do not formulate this as sequence-to-sequence learning. We use the shared recurrent sentence encoder to encode both the premise and hypothesis into fixed length vectors $u$ and $v$, respectively. We then feed the vector $[u; v; |u - v|; u * v]$, which is a concatenation of the premise and hypothesis vectors and their respective absolute difference and hadamard product, to an MLP that performs the 3-way classification. This is the same classification strategy adopted by Conneau et al. (2017). We train on a collection of about 1 million sentence pairs from the SNLI (Bowman et al., 2015) and MultiNLI (Williams et al., 2017) corpora.

Table 1 presents a summary of the number of sentence pairs used for each task.

---

[1] `https://github.com/dlwh/puck`

| Task | Sentence Pairs |
|------|:--------------:|
| En-Fr (WMT14) | 40M |
| En-De (WMT15) | 5M |
| Skipthought (BookCorpus) | 74M |
| AllNLI (SNLI + MultiNLI) | 1M |
| Parsing (PTB + 1-billion word) | 4M |
| Total | 124M |

Table 1: An approximate number of sentence pairs for each task.

## 4.1 MULTI-TASK TRAINING SETUP

Multi-task training with different data sources for each task stills poses open questions. For example: When does one switch to training on a different task? Should the switching be periodic? Do we weight each task equally? If not, what training ratios do we use?

Dong et al. (2015) use periodic task alternations with equal training ratios for every task. In contrast, Luong et al. (2015) alter the training ratios for each task based on the size of their respective training sets. Specifically, the training ratio for a particular task, $\alpha_i$, is the fraction of the number of training examples in that task to the total number of training samples across all tasks. The authors then perform $\alpha_i * N$ parameter updates on task $i$ before selecting a new task at random proportional to the training ratios, where N is a predetermined constant.

We take a simpler approach and pick a new sequence-to-sequence task to train on after every parameter update sampled uniformly. An NLI minibatch is interspersed after every ten parameter updates on sequence-to-sequence tasks (this was chosen so as to complete roughly 6 epochs of the dataset after 7 days of training). Our approach is described formally in the Algorithm below.

Model details can be found in section 7 in the Appendix.

---

**Require:** A set of $k$ tasks with a common source language, a shared encoder $\mathbf{E}$ across all tasks and a set of $k$ task specific decoders $\mathbf{D_1} \ldots \mathbf{D_k}$. Let $\theta$ denote each model's parameters, $\alpha$ a probability vector $(p_1 \ldots p_k)$ denoting the probability of sampling a task such that $\Sigma_i^k p_i = 1$, datasets for each task $\mathbb{P}_1 \ldots \mathbb{P}_k$ and a loss function $L$.

---

**while** $\theta$ *has not converged* **do**

    1: Sample task $i \sim \mathbf{Cat}(k, \alpha)$.
    2: Sample input, output pairs $\mathbf{x}, \mathbf{y} \sim \mathbb{P}_i$.
    3: Input representation $\mathbf{h}_x \leftarrow \mathbf{E}_\theta(\mathbf{x})$.
    4: Prediction $\tilde{\mathbf{y}} \leftarrow \mathbf{D}_{i_\theta}(\mathbf{h}_x)$
    5: $\theta \leftarrow \text{Adam}(\nabla_\theta L(\mathbf{y}, \tilde{\mathbf{y}}))$.

**end**

---

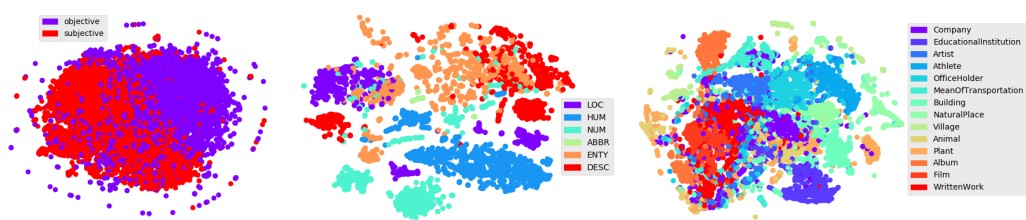

Figure 1: T-SNE visualizations of our sentence representations on 3 different datasets. SUBJ (left), TREC (middle), DBpedia (right). Dataset details are presented in the Appendix.

# 5 EVALUATION STRATEGIES, EXPERIMENTAL RESULTS & DISCUSSION

In this section, we describe our approach to evaluate the quality of our learned representations, present the results of our evaluation and discuss our findings.

## 5.1 EVALUATION STRATEGY

We follow a similar evaluation protocol to those presented in Kiros et al. (2015); Hill et al. (2016); Conneau et al. (2017) which is to use our learned representations as features for a low complexity classifier (typically linear) on a novel supervised task/domain unseen during training *without updating the parameters of our sentence representation model*. We also consider such a transfer learning evaluation in an artificially constructed low-resource setting. In addition, we also evaluate the quality of our learned individual word representations using standard benchmarks (Faruqui & Dyer, 2014; Tsvetkov et al., 2015).

The choice of transfer tasks and evaluation framework[2] are borrowed largely from Conneau et al. (2017). We provide a condensed summary of the tasks in section 10 in the Appendix but refer readers to their paper for a more detailed description.

## 5.2 EXPERIMENTAL RESULTS & DISCUSSION

Table 2 presents the results of training logistic regression on 10 different supervised transfer tasks using different fixed-length sentence representation. Supervised approaches trained from scratch on some of these tasks are also presented for comparison. We present performance ablations when adding more tasks and increasing the number of hidden units in our GRU (+L). Ablation specifics are presented in section 9 of the Appendix.

It is evident from Table 2 that adding more tasks improves the transfer performance of our model. Increasing the capacity our sentence encoder with more hidden units (+L) as well as an additional layer (+2L) also lead to improved transfer performance. We observe gains of 1.1-2.3% on the sentiment classification tasks (MR, CR, SUBJ & MPQA) over Infersent and smaller improvement of 0.5% on SST. We demonstrate substantial gains on TREC (6% over Infersent and roughly 2% over the CNN-LSTM), outperforming even a competitive supervised baseline. We see similar gains (2.3%) on paraphrase identification (MPRC), closing the gap on supervised approaches trained from scratch. The addition of constituency parsing improves performance on sentence relatedness (SICK-R) and entailment (SICK-E) consistent with observations made by Bowman et al. (2016).

In Table 4, we show that simply training an MLP on top of our fixed sentence representations outperforms several strong & complex supervised approaches that use attention mechanisms, even on this fairly large dataset. For example, we observe a 0.2-0.5% improvement over the decomposable attention model (Parikh et al., 2016). When using only a small fraction of the training data, indicated by the columns 1k-25k, we are able to outperform the Siamese and Multi-Perspective CNN using roughly 6% of the available training set. We also outperform the Deconv LVM model proposed by Shen et al. (2017) in this low-resource setting.

Unlike Conneau et al. (2017), who use pretrained GloVe word embeddings, we learn our word embeddings from scratch. Somewhat surprisingly, in Table 3 we observe that the learned word embeddings are competitive with popular methods such as GloVe, word2vec, and fasttext (Bojanowski et al., 2016) on the benchmarks presented by Faruqui & Dyer (2014) and Tsvetkov et al. (2015).

In Table 5, we probe our sentence representations to determine if certain sentence characteristics and syntactic properties can be inferred following work by Adi et al. (2016) and Shi et al. (2016). We observe that syntactic properties are better encoded with the addition of multi-lingual NMT and parsing. Representations learned solely from NLI do appear to encode syntax but incorporation into our multi-task framework does not amplify this signal. Similarly, we observe that sentence characteristics such as length and word order are better encoded with the addition of parsing.

In Appendix Table 6, we note that our sentence representations outperform skip-thoughts and are on par with Infersent for image-caption retrieval. We also observe that comparing sentences using

---

[2]`https://www.github.com/facebookresearch/SentEval`
[4]`https://github.com/kudkudak/word-embeddings-benchmarks/wiki`

| Model | MR | CR | SUBJ | MPQA | SST | TREC | MRPC | SICK-R | SICK-E | STSB | Δ |
|---|---|---|---|---|---|---|---|---|---|---|---|
| *Transfer approaches* | | | | | | | | | | | |
| FastSent | 70.8 | 78.4 | 88.7 | 80.6 | - | 76.8 | 72.2/80.3 | - | - | - | - |
| FastSent+AE | 71.8 | 76.7 | 88.8 | 81.5 | - | 80.4 | 71.2/79.1 | - | - | - | - |
| NMT En-Fr | 64.7 | 70.1 | 84.9 | 81.5 | - | 82.8 | - | - | - | - | - |
| CNN-LSTM | 77.8 | 82.1 | 93.6 | 89.4 | - | 92.6 | 76.5/83.8 | 0.862 | - | - | - |
| Skipthought | 76.5 | 80.1 | 93.6 | 87.1 | 82.0 | 92.2 | 73.0/82.0 | 0.858 | 82.3 | - | - |
| Skipthought + LN | 79.4 | 83.1 | 93.7 | 89.3 | 82.9 | 88.4 | - | 0.858 | 79.5 | 72.1/70.2 | - |
| Word Embedding Average | - | - | - | - | 82.2 | - | - | 0.860 | 84.6 | - | - |
| DiscSent + BiGRU | - | - | 88.6 | - | - | 81.0 | 71.6/- | - | - | - | - |
| DiscSent + unigram | - | - | 92.7 | - | - | 87.9 | 72.5/- | - | - | - | - |
| DiscSent + embed | - | - | 93.0 | - | - | 87.2 | 75.0/- | - | - | - | - |
| Byte mLSTM | **86.9** | **91.4** | **94.6** | 88.5 | - | - | 75.0/82.8 | 0.792 | - | - | - |
| Infersent (SST) | (*) | 83.7 | 90.2 | 89.5 | (*) | 86.0 | 72.7/80.9 | 0.863 | 83.1 | - | - |
| Infersent (SNLI) | 79.9 | 84.6 | 92.1 | 89.8 | 83.3 | 88. | 75.1/82.3 | _0.885_ | 86.3 | - | - |
| Infersent (AllNLI) | 81.1 | 86.3 | 92.4 | _90.2_ | **84.6** | 88.2 | 76.2/83.1 | 0.884 | _86.3_ | _75.8/75.5_ | 0.0 |
| *Our Models* | | | | | | | | | | | |
| +STN | 78.9 | 85.8 | 93.7 | 87.2 | 80.4 | 84.2 | 72.4/81.6 | 0.840 | 82.1 | 72.9/72.4 | -2.56 |
| +STN +Fr +De | 80.3 | 85.1 | 93.5 | 90.1 | 83.3 | 92.6 | 77.1/83.3 | 0.864 | 84.8 | 77.1/77.1 | 0.01 |
| +STN +Fr +De +NLI | 81.2 | 86.4 | 93.4 | 90.8 | 84.0 | 93.2 | 76.6/82.7 | 0.884 | 87.0 | 79.2/79.1 | 0.99 |
| +STN +Fr +De +NLI +L | 81.7 | 87.3 | _94.2_ | 90.8 | 84.0 | _94.2_ | 77.1/83.0 | 0.887 | 87.1 | 78.7/78.2 | 1.33 |
| +STN +Fr +De +NLI +L +STP | 82.7 | 88.0 | 94.1 | 91.2 | _84.5_ | 92.4 | 77.8/83.9 | 0.885 | 86.8 | 78.7/78.4 | 1.44 |
| +STN +Fr +De +NLI +2L +STP | _82.8_ | _88.3_ | 94.0 | **91.3** | 83.6 | 92.6 | 77.4/83.3 | 0.884 | 87.6 | **79.2/79.1** | 1.47 |
| +STN +Fr +De +NLI +L +STP +Par | 82.5 | 87.7 | 94.0 | 90.9 | 83.2 | 93.0 | **78.6/84.4** | **0.888** | **87.8** | 78.9/78.6 | **1.48** |
| *Approaches trained from scratch on these tasks* | | | | | | | | | | | |
| Naive Bayes SVM | 79.4 | 81.8 | 93.2 | 86.3 | 83.1 | - | - | - | - | - | |
| AdaSent | 83.1 | 86.3 | 95.5 | 93.3 | - | 92.4 | - | - | - | - | |
| TF-KLD | - | - | - | - | - | - | 80.4/85.9 | - | - | - | |
| Illinois LH | - | - | - | - | - | - | - | - | 84.5 | - | |
| Dependency tree LSTM | - | - | - | - | - | - | - | 0.868 | - | - | |
| Neural Semantic Encoder | - | - | - | - | 89.7 | - | - | - | - | - | |
| BLSTM-2DCNN | 82.3 | - | 94.0 | - | 89.5 | 96.1 | - | - | - | - | |

Table 2: Evaluation of sentence representations on a set of 10 tasks using a linear model trained using each model's representations. The FastSent and NMT En-Fr models are described in Hill et al. (2016), CNN-LSTM in Gan et al. (2016), Skipthought in Kiros et al. (2015), Word embedding average in Arora et al. (2016), DiscSent from Jernite et al. (2017), Byte mLSTM from Radford et al. (2017), Infersent in Conneau et al. (2017), Neural Semantic Encoder from Munkhdalai & Yu (2017), BLSTM-2DCNN from Zhou et al. (2016). STN, Fr, De, NLI, L, 2L, STP & Par stand for skip-thought next, French translation, German translation, natural language inference, large model, 2-layer large model, skip-thought previous and parsing respectively. Δ indicates the average improvement over Infersent (AllNLI) across all 10 tasks. For MRPC and STSB we consider only the F1 score and Spearman correlations respectively and we also multiply the SICK-R scores by 100 to have all differences in the same scale. Bold numbers indicate the best performing transfer model on a given task. Underlines are used for each task to indicate both our best performing model as well as the best performing transfer model that isn't ours.

cosine similarities correlates reasonably well with their relatedness on semantic textual similarity benchmarks (Appendix Table 7).

We also present qualitative analysis of our learned representations by visualizations using dimensionality reduction techniques (Figure 1) and nearest neighbor exploration (Appendix Table 8). Figure 1 shows t-sne plots of our sentence representations on three different datasets - SUBJ, TREC and DBpedia. DBpedia is a large corpus of sentences from Wikipedia labeled by category and used by Zhang et al. (2015). Sentences appear to cluster reasonably well according to their labels. The clustering also appears better than that demonstrated in Figure 2 of Kiros et al. (2015) on TREC and SUBJ. Appendix Table 8 contains sentences from the BookCorpus and their nearest neighbors. Sentences with some lexical overlap and similar discourse structure appear to be clustered together.

| Embedding | Dim | V143 | SIMLEX | WS353 | YP130 | MTurk771 | RG65 | MEN | QVEC |
|---|---|---|---|---|---|---|---|---|---|
| SENNA | 50 | 0.36 | 0.27 | 0.49 | 0.16 | 0.50 | 0.50 | 0.57 | - |
| NMT En-Fr | 620 | - | 0.46 | 0.49 | - | 0.46 | 0.59 | 0.49 | - |
| GloVe6B | 300 | 0.31 | 0.37 | 0.61 | 0.56 | 0.65 | 0.77 | 0.74 | 0.47 |
| GloVe840B | 300 | 0.34 | 0.41 | 0.71 | 0.57 | 0.71 | 0.76 | **0.80** | - |
| Skipgram GoogleNews | 300 | - | 0.44 | 0.70 | - | 0.67 | 0.76 | 0.74 | - |
| FastText | 300 | 0.40 | 0.38 | **0.74** | 0.53 | 0.67 | **0.80** | 0.76 | 0.50 |
| Multilingual | 512 | 0.26 | 0.40 | 0.68 | 0.42 | 0.60 | 0.65 | 0.76 | - |
| Charagram | 300 | - | 0.71 | - | - | - | - | - | - |
| Attract-Repel | 300 | - | 0.75 | - | - | - | - | - | - |
| *Our Model* | | | | | | | | | |
| +STN +Fr +De +NLI +L | 512 | **0.56** | **0.59** | 0.66 | **0.62** | 0.70 | 0.65 | 0.67 | **0.54** |

Table 3: Evaluation of word embeddings. All results were computed using Faruqui & Dyer (2014) with the exception of the Skipgram, NMT, Charagram and Attract-Repel embeddings. Skipgram and NMT results were obtained from Jastrzebski et al. (2017)[4]. Charagram and Attract-Repel results were taken from Wieting et al. (2016) and Mrkšić et al. (2017) respectively. We also report QVEC benchmarks (Tsvetkov et al., 2015)

| Model | Accuracy | | | | |
|---|---|---|---|---|---|
| | 1k | 5k | 10k | 25k | All (400k) |
| Siamese-CNN | - | - | - | - | 79.60 |
| Multi-Perspective-CNN | - | - | - | - | 81.38 |
| Siamese-LSTM | - | - | - | - | 82.58 |
| Multi-Perspective-LSTM | - | - | - | - | 83.21 |
| L.D.C | - | - | - | - | 85.55 |
| BiMPM | - | - | - | - | 88.17 |
| DecAtt (GloVe) | - | - | - | - | 86.52 |
| DecAtt (Char) | - | - | - | - | 86.84 |
| pt-DecAtt (Word) | - | - | - | - | 87.54 |
| pt-DecAtt (Char) | - | - | - | - | **88.40** |
| CNN (Random) | 56.3 | 59.2 | 63.8 | 68.9 | - |
| CNN (GloVe) | 58.5 | 62.4 | 66.1 | 70.2 | - |
| LSTM-AE | 59.3 | 63.8 | 67.2 | 70.9 | - |
| Deconv-AE | 60.2 | 65.1 | 67.7 | 71.6 | - |
| Deconv LVM | 62.9 | 67.6 | 69.0 | 72.4 | - |
| Infersent (LogReg) | 68.8 | 73.8 | 75.7 | 76.4 | - |
| Infersent (MLP) | 71.8 | 75.7 | 77.2 | 78.9 | 84.79 |
| *Our Models* | | | | | |
| +STN +Fr +De +NLI +L +STP (LogReg) | 70.4 | 74.8 | 75.9 | 77.2 | - |
| +STN +Fr +De +NLI +L +STP (MLP) | **74.7** | **77.7** | **78.3** | **81.4** | 87.01 |

Table 4: Supervised & low-resource classification accuracies on the Quora duplicate question dataset. Accuracies are reported corresponding to the number of training examples used. The first 6 rows are taken from Wang et al. (2017), the next 4 are from Tomar et al. (2017), the next 5 from Shen et al. (2017) and The last 4 rows are our experiments using Infersent (Conneau et al., 2017) and our models.

## 6   CONCLUSION & FUTURE WORK

We present a multi-task framework for learning general-purpose fixed-length sentence representations. Our primary motivation is to encapsulate the inductive biases of several diverse training signals used to learn sentence representations into a single model. Our multi-task framework includes a combination of sequence-to-sequence tasks such as multi-lingual NMT, constituency parsing and skip-thought vectors as well as a classification task - natural language inference. We demonstrate that the learned representations yield competitive or superior results to previous general-purpose sentence representation methods. We also observe that this approach produces good word embeddings.

| Model | Length | Content | Order | Passive | Tense | TSS |
|---|---|---|---|---|---|---|
| | *Sentence characteristics* | | | *Syntatic properties* | | |
| Majority Baseline | 22.0 | 50.0 | 50.0 | 81.3 | 77.1 | 56.0 |
| Infersent (AllNLI) | 75.8 | **75.8** | 75.1 | 92.1 | 93.3 | 70.4 |
| Skipthought | - | - | - | 94.0 | 96.5 | 74.7 |
| *Our Models* | | | | | | |
| Skipthought +Fr +De | 86.8 | 75.7 | 81.1 | 97.0 | 97.0 | 77.1 |
| Skipthought +Fr +De +NLI | 88.1 | 75.7 | 81.5 | 96.7 | 96.8 | 77.1 |
| Skipthought +Fr +De +NLI +Par | **93.7** | 75.5 | **83.1** | **98.0** | **97.6** | **80.7** |

Table 5: Evaluation of sentence representations by probing for certain sentence characteristics and syntactic properties. Sentence length, word content & word order from Adi et al. (2016) and sentence active/passive, tense and top level syntactic sequence (TSS) from Shi et al. (2016). Numbers reported are the accuracy with which the models were able to predict certain characteristics.

In future work, we would like understand and interpret the inductive biases that our model learns and observe how it changes with the addition of different tasks beyond just our simple analysis of sentence characteristics and syntax. Having a rich, continuous sentence representation space could allow the application of state-of-the-art generative models of images such as that of Nguyen et al. (2016) to language. One could also consider controllable text generation by directly manipulating the sentence representations and realizing it by decoding with a conditional language model.

## ACKNOWLEDGEMENTS

The authors would like to thank Chinnadhurai Sankar, Sebastian Ruder, Eric Yuan, Tong Wang, Alessandro Sordoni, Guillaume Lample and Varsha Embar for useful discussions. We are also grateful to the PyTorch development team (Paszke et al., 2017). We thank NVIDIA for donating a DGX-1 computer used in this work and Fonds de recherche du Québec - Nature et technologies for funding.

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

APPENDIX

## 7 MODEL TRAINING

We present some architectural specifics and training details of our multi-task framework. Our shared encoder uses a common word embedding lookup table and GRU. We experiment with unidirectional, bidirectional and 2 layer bidirectional GRUs (details in Appendix section 9). For each task, every decoder has its separate word embedding lookups, conditional GRUs and fully connected layers that project the GRU hidden states to the target vocabularies. The last hidden state of the encoder is used as the initial hidden state of the decoder and is also presented as input to all the gates of the GRU at every time step. For natural language inference, the same encoder is used to encode both the premise and hypothesis and a concatenation of their representations along with the absolute difference and hadamard product (as described in Conneau et al. (2017)) are given to a single layer MLP with a dropout (Srivastava et al., 2014) rate of 0.3. All models use word embeddings of 512 dimensions and GRUs with either 1500 or 2048 hidden units. We used minibatches of 48 examples and the Adam Kingma & Ba (2014) optimizer with a learning rate of 0.002. Models were trained for 7 days on an Nvidia Tesla P100-SXM2-16GB GPU. While Kiros et al. (2015) report close to a month of training, we only train for 7 days, made possible by advancements in GPU hardware and software (cuDNN RNNs).

We did not tune any of the architectural details and hyperparameters owing to the fact that we were unable to identify any clear criterion on which to tune them. Gains in performance on a specific task do not often translate to better transfer performance.

## 8 VOCABULARY EXPANSION & REPRESENTATION POOLING

In addition to performing 10-fold cross-validation to determine the L2 regularization penalty on the logistic regression models, we also tune the way in which our sentence representations are generated

from the hidden states corresponding to words in a sentence. For example, Kiros et al. (2015) use the last hidden state while Conneau et al. (2017) perform max-pooling across all of the hidden states. We consider both of these approaches and pick the one with better performance on the validation set. We note that max-pooling works best on sentiment tasks such as MR, CR, SUBJ and MPQA, while the last hidden state works better on all other tasks.

We also employ vocabulary expansion on all tasks as in Kiros et al. (2015) by training a linear regression to map from the space of pre-trained word embeddings (GloVe) to our model's word embeddings.

## 9 MULTI-TASK MODEL DETAILS

This section describes the specifics our multi-task ablations in the experiments section. These definitions hold for all tables except for 3 and 5. We refer to skip-thought next as STN, French and German NMT as Fr and De, natural language inference as NLI, skip-thought previous as STP and parsing as Par.

**+STN +Fr +De** : The sentence representation $h_x$ is the concatenation of the final hidden vectors from a forward GRU with 1500-dimensional hidden vectors and a bidirectional GRU, also with 1500-dimensional hidden vectors.

**+STN +Fr +De +NLI** : The sentence representation $h_x$ is the concatenation of the final hidden vectors from a bidirectional GRU with 1500-dimensional hidden vectors and another bidirectional GRU with 1500-dimensional hidden vectors trained without NLI.

**+STN +Fr +De +NLI +L** : The sentence representation $h_x$ is the concatenation of the final hidden vectors from a bidirectional GRU with 2048-dimensional hidden vectors and another bidirectional GRU with 2048-dimensional hidden vectors trained without NLI.

**+STN +Fr +De +NLI +L +STP** : The sentence representation $h_x$ is the concatenation of the final hidden vectors from a bidirectional GRU with 2048-dimensional hidden vectors and another bidirectional GRU with 2048-dimensional hidden vectors trained without STP.

**+STN +Fr +De +NLI +2L +STP** : The sentence representation $h_x$ is the concatenation of the final hidden vectors from a 2-layer bidirectional GRU with 2048-dimensional hidden vectors and a 1-layer bidirectional GRU with 2048-dimensional hidden vectors trained without STP.

**+STN +Fr +De +NLI +L +STP +Par** : The sentence representation $h_x$ is the concatenation of the final hidden vectors from a bidirectional GRU with 2048-dimensional hidden vectors and another bidirectional GRU with 2048-dimensional hidden vectors trained without Par.

In tables 3 and 5 we do not concatenate the representations of multiple models.

## 10 DESCRIPTION OF EVALUATION TASKS

Kiros et al. (2015) and Conneau et al. (2017) provide a detailed description of tasks that are typically used to evaluate sentence representations. We provide a condensed summary and refer readers to their work for a more thorough description.

### 10.1 TEXT CLASSIFICATION

We evaluate on text classification benchmarks - sentiment classification on movie reviews (MR), product reviews (CR) and Stanford sentiment (SST), question type classification (TREC), subjectivity/objectivity classification (SUBJ) and opinion polarity (MPQA). Representations are used to train a logistic regression classifier with 10-fold cross validation to tune the L2 weight penalty. The evaluation metric for all these tasks is classification accuracy.

### 10.2 PARAPHRASE IDENTIFICATION

We also evaluate on pairwise text classification tasks such as paraphrase identification on the Microsoft Research Paraphrase Corpus (MRPC) corpus. This is a binary classification problem to

identify if two sentences are paraphrases of each other. The evaluation metric is classification accuracy and F1.

## 10.3 Entailment and Semantic Relatedness

To test if similar sentences share similar representations, we evaluate on the SICK relatedness (SICK-R) task where a linear model is trained to output a score from 1 to 5 indicating the relatedness of two sentences. We also evaluate using the entailment labels in the same dataset (SICK-E) which is a binary classification problem. The evaluation metric for SICK-R is Pearson correlation and classification accuracy for SICK-E.

## 10.4 Semantic Textual Similarity

In this evaluation, we measure the relatedness of two sentences using only the cosine similarity between their representations. We use the similarity textual similarity (STS) benchmark tasks from 2012-2016 (STS12, STS13, STS14, STS15, STS16, STSB). The STS dataset contains sentences from a diverse set of data sources. The evaluation criteria is Pearson correlation.

## 10.5 Image-caption retrieval

Image-caption retrieval is typically formulated as a ranking task wherein images are retrieved and ranked based on a textual description and vice-versa. We use 113k training images from MSCOCO with 5k images for validation and 5k for testing. Image features are extracted using a pre-trained 110 layer ResNet. The evaluation criterion is Recall@K and the median K across 5 different splits of the data.

## 10.6 Quora Duplicate Question Classification

In addition to the above tasks which were considered by Conneau et al. (2017), we also evaluate on the recently published Quora duplicate question dataset[5] since it is an order of magnitude larger than the others (approximately 400,000 question pairs). The task is to correctly identify question pairs that are duplicates of one another, which we formulate as a binary classification problem. We use the same data splits as in Wang et al. (2017). Given the size of this data, we consider a more expressive classifier on top of the representations of both questions. Specifically, we train a 4 layer MLP with 1024 hidden units, with a dropout rate of 0.5 after every hidden layer. The evaluation criterion is classification accuracy. We also artificially create a low-resource setting by reducing the number of training examples between 1,000 and 25,000 using the same splits as Shen et al. (2017).

## 10.7 Sentence Characteristics & Syntax

In an attempt to understand what information is encoded in by sentence representations, we consider six different classification tasks where the objective is to predict sentence characteristics such as length, word content and word order (Adi et al., 2016) or syntactic properties such as active/passive, tense and the top syntactic sequence (TSS) from the parse tree of a sentence (Shi et al., 2016).

The sentence characteristic tasks are setup in the same way as described in Adi et al. (2016). The length task is an 8-way classification problem where sentence lengths are binned into 8 ranges. The content task is formulated as a binary classification problem that takes a concatenation of a sentence representation $u \in \mathbb{R}^k$ and a word representation $w \in \mathbb{R}^d$ to determine if the word is contained in the sentence. The order task is an extension of the content task where a concatenation of the sentence representation and word representations of two words in sentence is used to determine if the first word occurs before or after the second. We use a random subset of the 1-billion-word dataset for these experiments that were not used to train our multi-task representations.

The syntactic properties tasks are setup in the same way as described in Shi et al. (2016).The passive and tense tasks are characterized as binary classification problems given a sentence's representation. The former's objective is to determine if a sentence is written in active/passive voice while the latter's objective is to determine if the sentence is in the past tense or not. The top syntactic sequence

---

[5]https://data.quora.com/First-Quora-Dataset-Release-Question-Pairs

(TSS) is a 20-way classification problem with 19 most frequent top syntactic sequences and 1 miscellaneous class. We use the same dataset as the authors but different training, validation and test splits.

| Model | Caption Retrieval | | | | Image Retrieval | | | |
|---|---|---|---|---|---|---|---|---|
| | R@1 | R@5 | R@10 | Med r | R@1 | R@5 | R@10 | Med r |
| *Sentence Representations trained from scratch* | | | | | | | | |
| m-CNN | 38.3 | - | 81.0 | 2 | 27.4 | - | 79.5 | 3 |
| m-CNN Ensemble | 42.8 | - | 84.1 | 2 | 32.6 | - | 82.8 | 3 |
| Oreder-embeddings | 46.7 | - | 88.9 | 2 | 37.9 | - | 85.9 | 2 |
| *Transfer learning approaches* | | | | | | | | |
| Skipthought | 37.9 | 72.2 | 84.3 | 2 | 30.6 | 66.2 | 81.0 | 3 |
| Infersent (SNLI) | 42.4 | **76.1** | 87.0 | 2 | 33.2 | 69.7 | 83.6 | 3 |
| Infersent (AllNLI) | 42.6 | 75.3 | **87.3** | 2 | **33.9** | 69.7 | **83.8** | 3 |
| +STN +Fr +De +NLI +L +STP | **43.0** | 76.0 | 87.0 | 2 | 33.8 | **70.1** | 83.6 | 2.8 |

Table 6: COCO Retrieval with ResNet-101 features

| Model | STS12 | STS13 | STS14 | STS15 | STS16 |
|---|---|---|---|---|---|
| *Unsupervised/Transfer Approaches* | | | | | |
| Skipthought+LN | 30.8 | 24.8 | 31.4 | 31.0 | - |
| GloVe average | 52.5 | 42.3 | 54.2 | 52.7 | - |
| GloVe TF-IDF | 58.7 | 52.1 | 63.8 | 60.6 | - |
| GloVe + WR (U) | 56.2 | 56.6 | 68.5 | 71.1 | - |
| Charagram-phrase | **66.1** | **57.2** | **74.7** | **76.1** | - |
| Infersent | 59.2 | 58.9 | 69.6 | 71.3 | **71.4** |
| +STN +Fr +De +NLI +L +STP | 60.6 | 54.7 | 65.8 | 74.2 | 66.4 |
| *Supervised Approaches* | | | | | |
| LSTM w/o output gates | 51.0 | 45.2 | 59.8 | 63.9 | - |
| PP-Proj | 60.0 | 56.8 | 71.3 | 74.8 | - |

Table 7: Evaluation of sentence representations on the semantic textual similarity benchmarks. Numbers reported are Pearson Correlations x100. Skipthought, GloVe average, GloVe TF-IDF, GloVe + WR (U) and all supervised numbers were taken from Arora et al. (2016) and Wieting et al. (2015) and Charagram-phrase numbers were taken from Wieting et al. (2016). Other numbers were obtained from the evaluation suite provided by Conneau et al. (2017)

| Query and nearest sentence |
|---|
| **i ... i want ... ” she could n't finish the sentence and she did n't have to .** |
| i ... ” she had no idea what she had intended to say . |
| and ... ” she could n't finish . |
| i ... ” he winced as if he could n't bear whatever flashed through his mind . |
| you were going to sleep with her just to ... ” i could n't even finish my sentence . |
| **tyler was kind enough to wait two weeks before talking about their relationship again .** |
| he 'd been more than honest in setting forth his feelings about their relationship . |
| she did n't wait for him to ask what she was talking about . |
| damn , just thinking about it made him want to talk to her again . |
| i did n't wait for my bestie to try and talk me out of it . |
| **“ you can , my lady . ”** |
| “ of course , my lady . ” |
| “ what can i do for you , my lady ? ” |
| “ watch yourself , my lady . ” |
| “ where to , my lady ? ” |
| **they 'd kept silent about it , a terrifying secret between them that was n't going away .** |
| he 'd known they had secrets between them , things they were n't ready to share . |
| she 'd kept his secret , even from her two closest confidantes . |
| he 'd made her a promise and he was n't about to break it . |
| she 'd brushed them off as shock-induced ramblings , but now riley was n't so sure . |
| **“ you deserve good things , ” marshal said , but he was still wearing that damn pitying smile .** |
| “ i 'm really happy to see you , ladybug , ” he said softly , the smile never leaving his eyes . |
| “ you 're a bad liar , ” he said , still smiling . |
| “ it 's okay , ” i said , but he shook his head and sighed . |
| “ i 'll answer that , ” he said , his eyes still lit with humor . |
| **belatedly i turned to look up at max and exclaimed , “ hey ! ”** |
| i started , hollering at his retreating form , “ thank you ! ” |
| he slowly turned and looked at me as he barely said , “ fine . ” |
| my head came up and i asked , “ sorry ? ” |
| i shook my head and mouthed , “ i ca n't believe you ! ” |
| **“ can you make a- ” marty , raegan 's regular , began .** |
| “ it 's not really a- ” i began , but lesley interrupted me . |
| “ can i set you up with someone- ” “ no , ” he barked . |
| “ i 'll raise it , ” schuld said . |
| “ you 're a mensch , ” he said . |

Table 8: A query sentence and its nearest neighbors sorted by decreasing cosine similarity using our model. Sentences and nearest neighbors were chosen from a random subset of 500,000 sentences from the BookCorpus

