# OpenReview forum: "Learning General Purpose Distributed Sentence Representations via Large Scale Multi-task Learning"
_ICLR.cc/2018/Conference — Accept (Poster)_

### Official Review · AnonReviewer3 · 2017-11-19
**Generally positive review**

**Rating:** 8
**Confidence:** 5

**Review:**

Follow-Up Comments
----

I continue to argue that this paper makes a contribution to a major open question, and clearly warrants acceptance.

I agree with R1 that the results do not tell a completely clear story, and that the benefits of pretraining are occasionally minimal or absent. However, R1 uses this as the basis to argue for rejection, which does not seem reasonable to me at all. This limitation is an empirical fact that the paper has done a reasonable job of revealing, and it does not take away the paper's reason for existence, since many of the results are still quite strong, and the trends do support the merit of the proposed approach.

The authors mostly addressed my main concern, which was the relatively weak ablation. More combinations would be nice, but assuming reasonable resource constraints, I think the authors have done their due diligence, and the paper makes a clear contribution. I disagree with the response, though, that the authors can lean on other papers to help fill in the ablation—every paper in this area uses subtly different configurations.

I have one small lingering concern, which is not big enough to warrant acceptance: R2's point 10 is valid—the use of multiple RNNs trained on different objectives in the ablation experiments unexpected and unusual, and deserves mention in the body of the paper, rather than only in an appendix.

----
Original Review
---

This paper explores a variety of tasks for the pretraining of a bidirectional GRU sentence encoder for use in data-poor downstream tasks. The authors find that the combination of supervised training with NLI, MT, and parsing, plus unsupervised training on the SkipThought objective yields a model that robustly outperforms the best prior method on every task included in the standard SentEval suite, and several others.

This paper isn't especially novel. The main results of the paper stem from a combination of a few ideas that were ripe for combination (SkipThought from Kiros, BiLSTM-max and S/MNLI from Conneau, MT from McCann, parsing following Luong, etc.). However, the problem that the paper addresses is a major open issue within NLP, and the paper is very well done, so it would be in the best interest of all involved to make sure that the results are published promptly. I strongly support acceptance.

My one major request would be a more complete ablation analysis. It would be valuable for researchers working on other languages (among others) to know which labeled or unlabeled datasets contributed the most. Your ablation does not offer enough evidence to one to infer this---among other things, NLI and MT are never presented in isolation, and parsing is never presented without those two. Minimally, this should involve presenting results for models trained separately on each of the pretraining tasks.

I'll also echo another question from Samuel's comment: Could you say more about how you conducted the evaluation on the SentEval tasks? Did your task-specific model (or the training/tuning procedure for that model) differ much from prior work?

Details:

The paragraph starting "we take a simpler approach" is a bit confusing. If task batches are sampled *uniformly*, how is NLI be sampled less often than the other tasks?

Given how many model runs are presented, and that the results don't uniformly favor your largest/last model, it'd be helpful to include some kind of average of performance across tasks that can be used as a single-number metric for comparison. This also applies to the word representation evaluation table.

When comparing word embeddings, it would be helpful to include the 840B-word release of GloVe embeddings. Impressionistically, that is much more widely used than the older 6B-word release for which Faruqui reports numbers. This isn't essential to the paper, but it would make your argument in that section more compelling.

"glove" => GloVe; "fasttext" => fastText

---

> ### Author Response · Authors · 2018-01-05
> **Clarifications - Part 1**
>
> Hi,
>
> Thank you for your reviews and positive feedback. We’ve drafted responses to your concerns below.
>
> (1) "My one major request would be a more complete ablation analysis. It would be valuable for researchers working on other languages (among others) to know which labeled or unlabeled datasets contributed the most. Your ablation does not offer enough evidence to one to infer this---among other things, NLI and MT are never presented in isolation, and parsing is never presented without those two. Minimally, this should involve presenting results for models trained separately on each of the pretraining tasks."
>
> (Response 1) Regarding your point and in response to Sam Bowman’s comment we ran a model with just skipthoughts whose results have been added to Table 2 of the revised manuscript. Experiments for other tasks in isolation have already been presented in prior work (ex: NLI - Conneau et al (2017), MT - Hill et al (2015), Skipthoughts - Kiros et al (2015)), the results from those experimental configurations are shown in Table 2. Analysing the results of prior work in Table 2 in the context of our experiments we can examine the impact of different tasks on the quality of the learned representations. For example the results of Conneau et al (2017) that use NLI outperform Skipthoughts (Kiros et al 2015) which in turn outperforms NMT En-Fr (Hill et al 2015). But as we note in our experiment (+STN +Fr +De), it is possible to combine the benefits of skipthoughts and NMT to yield comparable performance to NLI (delta of 0.01 from Infersent). Adding more tasks and increasing model capacity (to match Conneau et al) from this point appears to produce reasonable improvements. An even more thorough analysis of the impact of each task on the learned representations is definitely something we intend to focus on in future work.
>
> (2) "I'll also echo another question from Samuel's comment: Could you say more about how you conducted the evaluation on the SentEval tasks? Did your task-specific model (or the training/tuning procedure for that model) differ much from prior work?"
>
> (Response 2) We used the SentEval evaluation suite, as is, with a single change from the default 5-fold cross validation to 10-fold cross validation. This was to match the same setting used by Conneau et al (2017), following instructions on their GitHub page “kfold (int): k in the kfold-validation. Set to 10 to be comparable to published results (default: 5)”.
>
> We also tuned the way in which the fixed-length sentence representation is computed given the hidden states corresponding to each word, by either max-pooling all the hidden states or selecting the last hidden state.
>
> (3) "The paragraph starting "we take a simpler approach" is a bit confusing. If task batches are sampled *uniformly*, how is NLI be sampled less often than the other tasks?"
>
> (Response 3) All tasks except NLI are sampled uniformly.  We intersperse one NLI minibatch after 10 updates on other tasks as described. We have changed the description in the revised manuscript to indicate that we sample a new sequence-to-sequence task uniformly, making it clear that NLI is sampled less frequently.

---

> ### Author Response · Authors · 2018-01-05
> **Clarifications - Part 2**
>
> (4) "Given how many model runs are presented, and that the results don't uniformly favor your largest/last model, it'd be helpful to include some kind of average of performance across tasks that can be used as a single-number metric for comparison. This also applies to the word representation evaluation table."
>
> (Response 4) Thank you for suggesting this. To quantify transfer performance with a single number, we use the mean difference of our models from Infersent (AllNLI) in Table 2 across all 10 tasks (*). The results are : 0.01|0.99|1.33|1.39|1.47|1.74. As evident from the results, adding more tasks certainly seems to help on average, even in the parsing scenario (1.39 vs 1.47 for a model of same architecture and capacity); however, adding more capacity (+2L) seems to have a greater impact.
>
> However, since the tasks presented in Table 2 are reasonably diverse, one might build representations that are suited to small subsets of these tasks but not others (see for example Radford et al (2017), who achieve very impressive results on sentiment classification tasks such as MR/CR/SST, etc.). Having a single evaluation score across all tasks might not be meaningful in such cases.
>
> *  For MRPC and STSB we consider only the F1 score and Spearman scores respectively and we also multiply the SICK-R scores by 100 to map all differences to the same scale.
>
> (5) "When comparing word embeddings, it would be helpful to include the 840B-word release of GloVe embeddings. Impressionistically, that is much more widely used than the older 6B-word release for which Faruqui reports numbers. This isn't essential to the paper, but it would make your argument in that section more compelling."
>
> (Response 5) Thank you for the suggestion. The results for the glove 840B vectors are: |0.34|0.41|0.71|0.46|0.57|0.71|0.76|0.8|. Since this release contains case-sensitive vectors, we only pick vectors for the lower-cased words. These results have been included in the revised manuscript.

---

### Official Review · AnonReviewer1 · 2017-11-28

**Rating:** 4
**Confidence:** 5

**Review:**

This paper shows that learning sentence representations from a diverse set of tasks (skip-thought objective, MT, constituency parsing, and natural language inference) produces .
The main contribution of the paper is to show learning from multiple tasks improves the quality of the learned representations.
Experiments on various text classification and sentiment analysis datasets show that the proposed method is competitive with existing approaches.
There is an impressive number of experiments presented in the paper, but the results are a bit mixed, and it is not always clear that adding more tasks help.

I think this paper addresses an important problem of learning general purpose sentence representations.
However, I am unable to draw a definitive conclusion from the paper.
From Table 2, the best performing model is not always the one with more tasks.
For example, adding a parsing objective can either improve or lower the performance quite significantly.
Could it be that datasets such as MRPC, SICK, and STSB require more understanding of syntax?
Even if this is the case, why adding this objective hurt performance for other datasets?
Importantly, it is also not clear whether the performance improvement comes from having more unlabeled data (even if it is trained with the same training objective) or having multiple training objectives.
Another question I have is that if there is any specific reason that language modeling is not included as one of the training objectives to learn sentence representations, given that it seems to be the easiest one to collect training data for.

The results for transfer learning and low resource settings are more positive.
However, it is not surprising that pretraining parts of the model on a large amount of unlabeled data helps when there is not a lot of labeled examples.

Overall, while the main contribution of the paper is that having multiple training objectives help learning better sentence, I am not yet convinced by the experiments that this is indeed the case.

---

> ### Author Response · Authors · 2018-01-05
> **Clarifications regarding some concerns raised**
>
> Hi,
>
> Thank you for your reviews. We appreciate the feedback and constructive criticism. We’ve drafted some responses to your comments below.
>
> 1) "I think this paper addresses an important problem of learning general purpose sentence representations.
> However, I am unable to draw a definitive conclusion from the paper.
> From Table 2, the best performing model is not always the one with more tasks.
> For example, adding a parsing objective can either improve or lower the performance quite significantly.
> Could it be that datasets such as MRPC, SICK, and STSB require more understanding of syntax?
> Even if this is the case, why adding this objective hurt performance for other datasets?"
>
> (Response 1) In response to your concern about performance not strictly increasing with the addition of more tasks, we believe that there is no free lunch when it comes to which of these models to use.  As we argue in the motivation of the paper, there may not always be a single (or multiple) training objective that results in improvements across all possible transfer learning benchmarks. Different training objectives will result in different learned inductive biases which in turn will affect results on different transfer learning tasks. In future work, we hope to uncover some of these inductive biases in greater detail which we can use to understand the relationships between the tasks used to train our model and its performance on a particular transfer task.
>
> However, a closer look at the improvements obtained with the addition of more tasks in Table 2 reveals that adding more tasks does indeed help on average (across all 10 tasks). To quantify transfer performance with a single number, we use the mean difference of our models from Infersent (AllNLI) in Table 2 across all 10 tasks (*). The results are : 0.01|0.99|1.33|1.39|1.47|1.74. As evident from the results, adding more tasks certainly seems to help on average, even in the parsing scenario (1.39 vs 1.47 for a model of same architecture and capacity) however adding more capacity (+2L) seems to have a bigger impact.
>
> Regarding your concerns about not seeing improvements across all tasks when adding a new task, this could be because although we add more tasks, we do not increase model capacity. Increased capacity may be required to learn all tasks effectively, or it may be that the inductive biases learned with the addition of new tasks are not useful for some subset of tasks (as you point out regarding the “understanding of syntax” in MRPC, SICK and STSB).
>
> *  For MRPC and STSB we consider only the F1 score and Spearman scores respectively and we also multiply the SICK-R scores by 100 to map all differences to the same scale.
>
> (2) "Importantly, it is also not clear whether the performance improvement comes from having more unlabeled data (even if it is trained with the same training objective) or having multiple training objectives.
> Another question I have is that if there is any specific reason that language modeling is not included as one of the training objectives to learn sentence representations, given that it seems to be the easiest one to collect training data for."
>
> (Response 2) Since all models were run for a total of 7 days, those with the same capacity see the same number of training examples and also undergo the same number of parameter updates. Adding more tasks increases the diversity of data the model observes. The skip-thoughts experiment compared to the skip-thoughts + translation result illustrates this point particularly well, showing increased performance from adding the translation task which serves as a substitute for the alternative single task setup with additional skip-thought training data. Statistically speaking over even more experiments, as discussed above we observe a general trend that with a fixed training example budget the addition of new tasks generally improves performance.
>
> We did not train our models with an (unconditional) language modeling objective because we believe that this does not lend itself easily to learning fixed length sentence representations. For example, the hidden state corresponding to the last word in a sentence is very unlikely to capture the entire history of the sentence. Also, a teacher-forced language modeling objective also emphasizes one-step-ahead prediction which we felt was not very well suited to learning representations that capture aspects of the sentence as a whole.

---

### Official Review · AnonReviewer2 · 2017-11-28
**solid empirical contribution to sentence embedding learning**

**Rating:** 8
**Confidence:** 5

**Review:**

---- updates: ----

I had a ton of comments and concerns, and I think the authors did an admirable job in addressing them.  I think the paper represents a solid empirical contribution to this area and is worth publishing in ICLR.

---- original review follows: ----

This paper is about learning sentence embeddings by combining a bunch of training signals: predicting the next & previous sentences (skip-thought), predicting the sentence's translation, classifying entailment relationships between two sentences, and predicting the constituent parse of a sentence. This is a simple idea that combines a bunch of things from prior work into one framework and yields strong results, outperforming most prior work on most tasks.

I think this paper is impressive in how it scales up training to use so many tasks and such large training sets for each task.  That and its strong experimental results make it worthy of publication. It's not very surprising that adding more tasks and data improves performance on average across downstream tasks, but it is nice to see the experimental results in detail. While many people would think of this idea, few would have the resources and expertise necessary to do it justice.  I also like how the authors move beyond the standard sentence tasks to evaluate also on the Quora question duplicate task with different amounts of training data and also consider the sentence characteristic / syntactic property tasks.  It would be great if the authors could release their pretrained sentence representation model so that other researchers could use it.

I do have some nitpicks here and there with the presentation and exposition, and I am concerned that at times the paper appears to be minimizing its weaknesses, but I think these are things that can be addressed in the next revision. I understand that sometimes it's tempting to minimize one's weaknesses in order to get a paper accepted because the reviewers may not understand the area very well and may get hung up on the wrong things. I understand the area well and so all the feedback I offer below comes from a place of desiring this paper's publication while also desiring it to be as accurate and helpful for the community as possible.

Below I'll discuss my concerns with the experiments and description of the results.

Regarding the results in Table 2:

The results in Table 2 seem a little bit unstable, as it is unclear which setting to use for the classification tasks; maybe it depends on the kind of classification being performed. One model seems best for the sentiment tasks ("+2L +STP") while other models seem best for SUBJ and MPQA.  Adding parsing as a training task hurts performance on the sentence classification tasks while helping performance on the semantic tasks, as the authors note.  It is unclear which is the best general model.  In particular, when others write papers comparing to the results in this paper, which setting should they compare to?  It would be nice if the authors could discuss this.

The results reported for the CNN-LSTM of Gan et al. do not exactly match those of any single row from Gan et al, either v1 or v2 on arxiv or the published EMNLP version. How were those specific numbers selected?

The caption of Table 2 states "All results except ours are taken from Conneau et al. (2017)." However, Conneau et al (neither the latest arxiv version nor the published EMNLP version) does not include many of the results in the table, such as CNN-LSTM and DiscSent mentioned in the following sentence in the caption. Did the authors replicate the results of those methods themselves, or report them from other papers?

What does bold and underlining indicate in Table 2?  I couldn't find this explained anywhere.

At the bottom of Table 2, in the section with approaches trained from scratch on these tasks, I'd suggest including the 89.7 SST result of Munkhdalai and Yu (2017) and the 96.1 TREC result of Zhou et al. (2016) (as well as potentially other results from Zhou et al, since they report results on others of these datasets). The reason this is important is because readers may observe that the paper's new method achieves higher accuracies on SST and TREC than all other reported results and mistakenly think that the new method is SOTA on those tasks.  I'd also suggest adding the results from Radford et al. (2017) who report 86.9 on MR and 91.4 on CR.  For other results on these datasets, including stronger results in non-fixed-dimensional-sentence-embedding transfer settings, see results and references in McCann et al. (2017).  While the methods presented in this paper are better than prior work in learning general purpose, fixed-dimensional sentence embeddings, they still do not produce state-of-the-art results on that many of these tasks, if any.  I think this is important to note.

For all tasks for which there is additional training, there's a confound due to the dimensionality of the sentence embeddings across papers. Using higher-dimensional sentence embeddings leads to more parameters in the linear model being trained on the task data. So it is unclear if the increase in hidden units in rows with "+L" is improving the results because of providing more weights for the linear model or whether it is learning a better sentence representation.

The main sentence embedding results are in Table 2, and use the SentEval framework. However, not all tasks are included. The STS Benchmark results are included, which use an additional layer trained on the STS Benchmark training data just like the SICK tasks. But the other STS results, which use cosine similarity on the embedding space directly without any retraining, are only included in the appendix (in Table 7). The new approach does not do very well on those unsupervised tasks. On two years of data it is better than InferSent and on two years it is worse. Both are always worse than the charagram-phrase results of Wieting et al (2016a), which has 66.1 on 2012, 57.2 on 2013, 74.7 on 2014, and 76.1 on 2015.  Charagram-phrase trains on automatically-generated paraphrase phrase pairs, but these are generated automatically from parallel text, the same type of resource used in the "+Fr" and "+De" models proposed in this submission, so I think it should be considered as a comparable model.

The results in the bottom section of Table 7, reported from Arora et al (2016), were in turn copied from Wieting et al (2016b), so I think it would make sense to also cite Wieting et al (2016b) if those results are to be included. Also, it doesn't seem appropriate to designate those as "Supervised Approaches" as they only require parallel text, which is a subset of the resources required by the new model.

There are some other details in the appendix that I find concerning:

Section 8 describes how there is some task-specific tuning of which function to compute on the encoder to produce the sentence representation for the task.  This means that part of the improvement over prior work (especially skip-thought and InferSent) is likely due to this additional tuning. So I suppose to use these sentence representations in other tasks, this same kind of tuning would have to be done on a validation set for each task?  Doesn't that slightly weaken the point about having "general purpose" sentence representations?

Section 9 provides details about how the representations are created for different training settings. I am confused by the language here. For example, the first setting ("+STN +Fr +De") is described as "A concatenation of the representations trained on these tasks with a unidirectional and bidirectional GRU with 1500 hidden units each." I'm not able to parse this. I think the authors mean "The sentence representation h_x is the concatenation of the final hidden vectors from a forward GRU (with 1500-dimensional hidden vectors) and a bidirectional GRU (also with 1500-dimensional hidden vectors)". Is this correct?

Also in Sec 9: I found it surprising how each setting that adds a training task uses the concatenation of a representation with that task and one without that task. What is the motivation for doing this? This seems to me to be an important point that should be discussed in Section 3 or 4. And when doing this, are the concatenated representations always trained jointly from scratch with the special task only updating a subset of the parameters, or do you use the fixed pretrained sentence representation from the previous row and just concatenate it with the new one?  To be more concrete, if I want to get the encoder for the second setting ("+STN +Fr +De +NLI"), do I have to train two times or can I just train once?  That is, the train-once setting would correspond to only updating the NLI-specific representation parameters when training on NLI data; on other data, all parameters would be updated. The train-twice setting would first train a representation on "+STN +Fr +De", then set it aside, then train a separate representation on "+STN +Fr +De +NLI", then finally concatenate the two representations as my sentence representation.  Do you use train-once or train-twice?

Regarding the results in Table 3:

What do bold and underline indicate?

What are the embeddings corresponding to the row labeled "Multilingual"?

In the caption, I can't find footnote 4.

The caption includes the sentence "our embeddings have 1040 pairs out of 2034 for which atleast one of the words is OOV, so a comparison with other embeddings isn't fair on RW."  How were those pairs handled?  If they were excluded, then I think the authors should not report results on RW.  I suspect that most of the embeddings included in the table also have many OOVs in the RW dataset but still compute results on it using either an unknown word embedding or some baseline similarity of zero for pairs with an OOV. I think the authors should find some way (like one of those mentioned, or some other way) of computing similarity of those pairs with OOVs. It doesn't make much sense to me to omit pairs with OOVs.

There are much better embeddings on SimLex than the embeddings whose results are reported in the table. Wieting et al. (2016a) report SimLex correlation of 0.706 and Mrkšić et al. (2017) report 0.751.  I'd suggest adding the results of some stronger embeddings to better contextualize the embeddings obtained by the new method.  Some readers may mistakenly think that the embeddings are SOTA on SimLex since no stronger results are provided in the table.


The points below are more minor/specific:

Sec. 2:

In Sec. 2, the paper discusses its focus on fixed-length sentence representations to distinguish itself from other work that produces sentence representations that are not fixed-length. I feel the motivation for this is lacking. Why should we prefer a fixed-length representation of a sentence? For certain downstream applications, it might actually be easier for practitioners to use a representation that provides a representation for each position in a sentence (Melamud et al., 2016; Peters et al., 2017; McCann et al., 2017) rather than an opaque sentence representation. Some might argue that since sentences have different lengths, it would be appropriate for a sentence representation to have a length proportional to the length of the sentence.  I would suggest adding some motivation for the focus on fixed-length representations.

Sec. 4.1:

"We take a simpler approach and pick a new task to train on after every parameter update sampled uniformly. An NLI minibatch is interspersed after every ten parameter updates on sequence-to-sequence tasks"
These two sentences seem contradictory. Maybe in the first sentence "pick a new task" should be changed to "pick a new sequence-to-sequence task"?

Sec. 5.1:

typo: "updating the parameters our sentence" --> "updating the parameters of our sentence"

Sec. 5.2:

typo in Table 4 caption: "and The" --> ". The"

typo: "parsing improvements performance" --> "parsing improves performance"


In general, there are many missing citations for the tasks, datasets, and prior work on them. I understand that the authors are pasting in numbers from many places and just providing pointers to papers that provide more citation info, but I think this can lead to mis-attribution of methods. I would suggest including citations for all datasets/tasks and methods whose results are being reported.


References:

McCann, Bryan, James Bradbury, Caiming Xiong, and Richard Socher. "Learned in translation: Contextualized word vectors." CoRR 2017.

Melamud, Oren, Jacob Goldberger, and Ido Dagan. "context2vec: Learning Generic Context Embedding with Bidirectional LSTM." CoNLL 2016.

Mrkšić, Nikola, Ivan Vulić, Diarmuid Ó. Séaghdha, Ira Leviant, Roi Reichart, Milica Gašić, Anna Korhonen, and Steve Young. "Semantic Specialisation of Distributional Word Vector Spaces using Monolingual and Cross-Lingual Constraints." TACL 2017.

Munkhdalai, Tsendsuren, and Hong Yu. "Neural semantic encoders." EACL 2017.

Pagliardini, Matteo, Prakhar Gupta, and Martin Jaggi. "Unsupervised Learning of Sentence Embeddings using Compositional n-Gram Features." arXiv preprint arXiv:1703.02507 (2017).

Peters, Matthew E., Waleed Ammar, Chandra Bhagavatula, and Russell Power. "Semi-supervised sequence tagging with bidirectional language models." ACL 2017.

Radford, Alec, Rafal Jozefowicz, and Ilya Sutskever. "Learning to generate reviews and discovering sentiment." arXiv preprint arXiv:1704.01444 2017.

Wieting, John, Mohit Bansal, Kevin Gimpel, and Karen Livescu. "Charagram: Embedding words and sentences via character n-grams." EMNLP 2016a.

Wieting, John, Mohit Bansal, Kevin Gimpel, and Karen Livescu. "Towards universal paraphrastic sentence embeddings." ICLR 2016b.

Zhou, Peng, Zhenyu Qi, Suncong Zheng, Jiaming Xu, Hongyun Bao, and Bo Xu. "Text Classification Improved by Integrating Bidirectional LSTM with Two-dimensional Max Pooling." COLING 2016.

---

> ### Author Response · Authors · 2018-01-05
> **Clarifications - Part 1**
>
> Hi,
>
> Thank you for your thorough and in-depth review. We appreciate the feedback and constructive criticism. We’ve addressed some your concerns below.
>
> (1) “The results in Table 2 seem a little bit unstable, as it is unclear which setting to use for the classification tasks; maybe it depends on the kind of classification being performed. One model seems best for the sentiment tasks ("+2L +STP") while other models seem best for SUBJ and MPQA.  Adding parsing as a training task hurts performance on the sentence classification tasks while helping performance on the semantic tasks, as the authors note.  It is unclear which is the best general model.  In particular, when others write papers comparing to the results in this paper, which setting should they compare to?  It would be nice if the authors could discuss this.”
>
> (Response 1) Indeed, there is no free lunch when it comes to which of these models to pick. As we argue in the motivation of the paper, there may not always be a single (or multiple) training objective that results in improvements across all possible transfer learning benchmarks. To understand how the addition of tasks impacts performance quantitatively, below we report the average improvement across all 10 tasks from Table 2 over Infersent (AllNLI) for each row (for our models). The results are : 0.01|0.99|1.33|1.39|1.47|1.74 (*), where we have reversed the order of last two experiments to make this point clearer here and in the revised manuscript. The 1.74 score results from a model with more capacity. When capacity remains constant (as in the other experiments) we see from these results that adding more tasks helps on average, even for parsing (ex. 1.39 vs 1.47 for a model of same architecture). However, adding more capacity in the +2L experiment also helps. While measuring the average improvement over Infersent on all SentEval tasks makes sense in this work, it may not necessarily be a good metric for other approaches, such as the sentiment neuron work of Radford et al. that learns to encode some aspects of a sentence better than others.
>
> * For MRPC and STSB we consider only the F1 score and Spearman scores respectively and we also multiply the SICK-R scores by 100 to have all differences in the same scale.
>
> (2) “The results reported for the CNN-LSTM of Gan et al. do not exactly match those of any single row from Gan et al, either v1 or v2 on arxiv or the published EMNLP version. How were those specific numbers selected?”
>
> (Response 2) Thank you for pointing this out. The results reported for the CNN-LSTM model of Gan et al. correspond to the best of their combine+emb and combine models in their v1 arXiv submission. We made a small mistake when reporting their results for MPQA, which should be 89.1 instead of 89.0. We have updated this row with results from their latest arXiv revision.
>
> (3) “What does bold and underlining indicate in Table 2?  I couldn't find this explained anywhere.”
>
> (Response 3) We used the same underline and bold semantics as in Conneau et al (2017). Bold numbers indicate the best performing transfer model on a given task. Underlines are used for each task to indicate both our best performing model as well as the best performing transfer model that isn't ours. We’ve clarified this in our revised manuscript.

---

> ### Author Response · Authors · 2018-01-05
> **Clarifications - Part 2**
>
> (4) “At the bottom of Table 2, in the section with approaches trained from scratch on these tasks, I'd suggest including the 89.7 SST result of Munkhdalai and Yu (2017) and the 96.1 TREC result of Zhou et al. (2016) (as well as potentially other results from Zhou et al, since they report results on others of these datasets). The reason this is important is because readers may observe that the paper's new method achieves higher accuracies on SST and TREC than all other reported results and mistakenly think that the new method is SOTA on those tasks.  I'd also suggest adding the results from Radford et al. (2017) who report 86.9 on MR and 91.4 on CR.  For other results on these datasets, including stronger results in non-fixed-dimensional-sentence-embedding transfer settings, see results and references in McCann et al. (2017).  While the methods presented in this paper are better than prior work in learning general purpose, fixed-dimensional sentence embeddings, they still do not produce state-of-the-art results on that many of these tasks, if any.  I think this is important to note.”
>
> (Response 4) These are great suggestions, we will certainly add more prior work in the “Approaches trained from scratch on these tasks” section of Table 2. Aside from clearing up any confusion about our model being state-of-the-art on these tasks, this will also give readers a sense of the gap between transfer learning approaches and purely supervised, task-specifc models. We’ve edited our manuscript to add results from Munkhdalai and Yu (2017), Zhou et al. (2016) and Radford et al. (2017). Regarding comparisons with McCann et al. (2017), it is unclear where it fits. It is a transfer approach but uses a heavily parameterized neural network as it’s task specific classifier in contrast to this and previous work that has used linear classifiers.
>
> (5) “For all tasks for which there is additional training, there's a confound due to the dimensionality of the sentence embeddings across papers. Using higher-dimensional sentence embeddings leads to more parameters in the linear model being trained on the task data. So it is unclear if the increase in hidden units in rows with "+L" is improving the results because of providing more weights for the linear model or whether it is learning a better sentence representation. “
>
> (Response 5) We observed improvements when increasing the number of hidden units even on tasks that are not evaluated using parametric models, such as in the STS* tasks (Appendix Table 7). We considered two models trained on the same subset of tasks but with different GRU hidden state sizes (+STN +Fr +De +NLI vs +STN +Fr +De +NLI +L). The following were our results on the STS12/13/14/15/16 benchmarks (Appendix Table 7) in the format |small vs large|. |60.8 vs 61.2|53.5 vs 53.4|64 vs 65.3|73.4 vs 74.6|64.9 vs 66.2|. These results have been added to the paper.
>
> (6) “The main sentence embedding results are in Table 2, and use the SentEval framework. However, not all tasks are included. The STS Benchmark results are included, which use an additional layer trained on the STS Benchmark training data just like the SICK tasks. But the other STS results, which use cosine similarity on the embedding space directly without any retraining, are only included in the appendix (in Table 7). The new approach does not do very well on those unsupervised tasks. On two years of data it is better than InferSent and on two years it is worse. Both are always worse than the charagram-phrase results of Wieting et al (2016a), which has 66.1 on 2012, 57.2 on 2013, 74.7 on 2014, and 76.1 on 2015.  Charagram-phrase trains on automatically-generated paraphrase phrase pairs, but these are generated automatically from parallel text, the same type of resource used in the "+Fr" and "+De" models proposed in this submission, so I think it should be considered as a comparable model. “
>
> (Response 6) We believe that our model doesn’t do as well on the STS benchmarks since our sentence representations capture certain features about a sentence, like sentence length, word order, and others, which aren’t very informative for tasks like semantic similarity. Even with a simple parametric model such as logistic regression in Table 2, the model can learn to down-weight such features and up-weight the ones that are actually useful to determine semantic similarity (STSB). With the STS evaluations that use cosine similarities, sentence vectors are expected to be similar in “every aspect”, which may not be a realistic expectation.We’ve also added charagram to Table 7 in our revised manuscript.

---

> ### Author Response · Authors · 2018-01-05
> **Clarifications - Part 3**
>
> (7) “The results in the bottom section of Table 7, reported from Arora et al (2016), were in turn copied from Wieting et al (2016b), so I think it would make sense to also cite Wieting et al (2016b) if those results are to be included. Also, it doesn't seem appropriate to designate those as "Supervised Approaches" as they only require parallel text, which is a subset of the resources required by the new model. “
>
> (Response 7) Thank you for pointing this out. We will certainly cite Weiting et al (2016b). We designated those models as “Supervised Approaches” since they were marked as such in Arora et al (2016).
>
> (8) “Section 8 describes how there is some task-specific tuning of which function to compute on the encoder to produce the sentence representation for the task.  This means that part of the improvement over prior work (especially skip-thought and InferSent) is likely due to this additional tuning. So I suppose to use these sentence representations in other tasks, this same kind of tuning would have to be done on a validation set for each task?  Doesn't that slightly weaken the point about having "general purpose" sentence representations?”
>
> (Response 8) We only tune the way in which we compute the fixed length sentence representations given all of the GRU’s hidden states (i.e., by max-pooling or picking the last hidden state). The GRU itself is still “general purpose”. Moreover, there appears to be an clear trend to infer which tasks benefit from max-pooling (sentiment related) and which benefit from using the last hidden state (others).
>
> (9) “Section 9 provides details about how the representations are created for different training settings. I am confused by the language here. For example, the first setting ("+STN +Fr +De") is described as "A concatenation of the representations trained on these tasks with a unidirectional and bidirectional GRU with 1500 hidden units each." I'm not able to parse this. I think the authors mean "The sentence representation h_x is the concatenation of the final hidden vectors from a forward GRU (with 1500-dimensional hidden vectors) and a bidirectional GRU (also with 1500-dimensional hidden vectors)". Is this correct?”
>
> (Response 9) Yes, this is correct. We’ve clarified this in our revised manuscript according to your suggestion.
>
> (10) “Also in Sec 9: I found it surprising how each setting that adds a training task uses the concatenation of a representation with that task and one without that task. What is the motivation for doing this? This seems to me to be an important point that should be discussed in Section 3 or 4. And when doing this, are the concatenated representations always trained jointly from scratch with the special task only updating a subset of the parameters, or do you use the fixed pretrained sentence representation from the previous row and just concatenate it with the new one?  To be more concrete, if I want to get the encoder for the second setting ("+STN +Fr +De +NLI"), do I have to train two times or can I just train once?  That is, the train-once setting would correspond to only updating the NLI-specific representation parameters when training on NLI data; on other data, all parameters would be updated. The train-twice setting would first train a representation on "+STN +Fr +De", then set it aside, then train a separate representation on "+STN +Fr +De +NLI", then finally concatenate the two representations as my sentence representation.  Do you use train-once or train-twice? “
>
> (Response 10) Our approach corresponds to your description of “train-twice”. We adopted the concatenation strategy described in this paper since it requires training only 1 model per set of training objectives (except for the initial set of objectives: +STN +Fr +De).
>
> (11) “Regarding the results in Table 3: What do bold and underline indicate? What are the embeddings corresponding to the row labeled "Multilingual"?”
>
> (Response 11) Bold and underline have the same semantics as in Table 2. The Multilingual embeddings correspond to Faruqui & Dyer’s (2014b) “Improving Vector Space Word Representations Using Multilingual Correlation” (we missed citing this paper, thanks for pointing this out).

---

> ### Author Response · Authors · 2018-01-05
> **Clarifications - Part 4**
>
> (12) “The caption includes the sentence "our embeddings have 1040 pairs out of 2034 for which atleast one of the words is OOV, so a comparison with other embeddings isn't fair on RW."  How were those pairs handled?  If they were excluded, then I think the authors should not report results on RW.  I suspect that most of the embeddings included in the table also have many OOVs in the RW dataset but still compute results on it using either an unknown word embedding or some baseline similarity of zero for pairs with an OOV. I think the authors should find some way (like one of those mentioned, or some other way) of computing similarity of those pairs with OOVs. It doesn't make much sense to me to omit pairs with OOVs.”
>
> (Response 12) The evaluation suite provided by Faruqui & Dyer (wordvectors.org), which we use, simply excludes words that are OOV. We have removed the RW column in the revised manuscript.
> (13) “There are much better embeddings on SimLex than the embeddings whose results are reported in the table. Wieting et al. (2016a) report SimLex correlation of 0.706 and Mrkšić et al. (2017) report 0.751.  I'd suggest adding the results of some stronger embeddings to better contextualize the embeddings obtained by the new method.  Some readers may mistakenly think that the embeddings are SOTA on SimLex since no stronger results are provided in the table.”
>
> (Response 13) Thank you for the pointers! We’ve added results from Weiting et al. (2016a) and Mrkšić et al. (2017) to Table 3 in the revised manuscript.
>
> (14) “In Sec. 2, the paper discusses its focus on fixed-length sentence representations to distinguish itself from other work that produces sentence representations that are not fixed-length. I feel the motivation for this is lacking. Why should we prefer a fixed-length representation of a sentence? For certain downstream applications, it might actually be easier for practitioners to use a representation that provides a representation for each position in a sentence (Melamud et al., 2016; Peters et al., 2017; McCann et al., 2017) rather than an opaque sentence representation. Some might argue that since sentences have different lengths, it would be appropriate for a sentence representation to have a length proportional to the length of the sentence.  I would suggest adding some motivation for the focus on fixed-length representations.”
>
> (Response 14) A thorough analysis of the pros and cons of fixed-length versus variable length sentence representations is something we hope to explore in the future. Some of the advantages of fixed-length representations include (a) being able to compute a straightforward non-parametric similarity score between two sentences (such as in the STS* tasks) (b) easy to use simple task-specific classifiers on top of the features extracted by our RNN (in contrast to work by McCann et al (2017) that uses a heavily parameterized task-specific classifier) (c) easy to manipulate aspects of a sentence with gradient based optimization for controllable generation such as in Mueller et al (2017).
>
> Also, similar to McCann et al (2017), it is possible to represent a sentence using the all of the RNN’s hidden states instead of just the last as in this work. In the classic, attention free encoder-decoder paradigm the architecture encourages all the necessary information for subsequent tasks from a variable length input to be captured in a fixed length vector. Concatenation of the intermediate hidden unit representations is possible but would likely contain redundant information with respect to the final code layer.   However, in future work, we would like to compare our approach more directly with McCann et al (2017) using their proposed Bi-attentive classification network with all of the hidden states of our general purpose multi-task GRU instead of just the last.
>
> (15) "We take a simpler approach and pick a new task to train on after every parameter update sampled uniformly. An NLI minibatch is interspersed after every ten parameter updates on sequence-to-sequence tasks"
> These two sentences seem contradictory. Maybe in the first sentence "pick a new task" should be changed to "pick a new sequence-to-sequence task"?
>
> (Response 15) Thank you for the above suggestion and noticing typos, we’ve made the edits you suggested to clarify things.
>
> References
>
> Mueller, Jonas, David Gifford, and Tommi Jaakkola. "Sequence to better sequence: continuous revision of combinatorial structures." International Conference on Machine Learning. 2017.

---

### Public Comment · ~Samuel_R._Bowman1 · 2017-11-01
**Neat! A few questions...**

Cool results. I very much appreciated the analysis (and Adi et al. results)!

– Do you have any results (formal or informal) on either SkipThought alone or MT alone using your implementation? It's a bit odd that the ablation starts with three tasks.
– Do you actually use SentEval (Conneau's evaluation software toolkit), or your own implementation of the same set of evaluations? You claim that you borrowed your evaluation 'largely' from them.
– Do you have any impression of how important it was to perform vocabulary expansion (relative to using an UNK token and/or raw w2v vectors)?

---

> ### Public Comment · (anonymous) · 2017-11-24
> **Clarifications**
>
> Hi,
>
> Thank you for your questions!
>
> 1. We did not consider starting our ablations with just SkipThought or NMT since they’ve been explored individually by Kiros et al 2015 and Hill et al 2016 respectively. We however, ran an experiment with just a large skip-thought next model (+STN + L) (4096 dimensions). Our results of this new ablation on the set of tasks presented in Table 2 (in the same column order) are - 78.9|85.8|93.7|87.2|80.4|84.2|72.4/81.6|0.840|82.1|72.9/72.4 , which indicates that there is some improvement with the addition of NMT even on a smaller model. We'll include these results in our first paper revision.
>
> 2. We did use SentEval in all evaluations except on the Quora dataset and Table 5 since they aren’t a part of SentEval (hence ‘largely’). We’ll make this clearer.
>
> 3. The following are the results for our model (+STN +Fr +De +NLI +L +STP) using vocabulary expansion with the glove 840B vectors versus using our learned <unk> token for every OOV on the set of tasks presented in Table 2. The results are in the same column order and are in format (vocab expansion/<unk> token). 82.2/81.83| 87.8/87.3| 93.8/93.7| 91.3/91.1| 84.5/84.0| 92.4/91.8| (78.0/83.8)/(78.4/84.1)| 0.885/0.884| 86.8/86.9| (79.2/78.8)/(79.0/78.6)|. These results indicate that there is a small benefit to performing vocabulary expansion.

---

> > ### Author Response · Authors · 2018-01-04
> > **Clarification regarding the comment above**
> >
> > Just wanted to point out that the above comment was made by the authors but doesn't show up as such since we commented from an alternate openreview ID that wasn't associated with this paper.

---

### Public Comment · ~Kyunghyun_Cho1 · 2017-11-28
**how do you know when to end learning?**

the models in this paper were trained for some arbitrary duration of 7 days. how was this duration selected, and how stable are the reported results w.r.t. different # of training days? if the training was terminated after 5 days (or 8 days or whatever that is not 7 days), would the results stay as they are reported in the submission?

---

> ### Author Response · Authors · 2018-01-05
> **A small experiment training one of our models for 5 more days**
>
> Hi,
>
> Thanks for the question! We did not observe strict improvements on our transfer tasks when training for longer, nor were perplexities on our validation sets strongly correlated with transfer performance.
>
> For example, when running our +STN +Fr +De +NLI +L +STP +Par model for 5 more days, we observed some (but not substantial) difference in transfer performance. To quantify transfer performance with a single number, we use the mean difference of our model from Infersent (AllNLI) in Table 2 across all 10 tasks (*). Our results on this metric for additional days of computation, starting at day (7) were: |(7) 1.47|(8) 1.21|(9) 1.36|(10) 1.52|(11) 1.53|(12) 1.37| with a mean of 1.41 and variance of 0.014.
>
> To determine if these changes were statistically significant, we generated two sets of scores, the first being day (x - Infersent) and the other being (day_(x + 1) - Infersent) (across all 10 tasks, over the additional 5 days) and then used a pairwise t-test between these two sets of scores. We found the p-values to be statistically insignificant from days 8 to 12.
>
> *  For MRPC and STSB we consider only the F1 score and Spearman scores respectively and we also multiply the SICK-R scores by 100 to map all differences to the same scale.

---

### Public Comment · (anonymous) · 2017-11-30
**size of embeddings**

In the Appendix, it is said "In tables 3 and 5 we do not concatenate the representations of multiple models.", which is a bit confusing. Are the embeddings a concatenation of separate encoders for the results in Table 2, and from a shared encoder in table 3 and 5?
It would be nice to include the size of the embeddings in Table 2 for a clearer and fair comparison to other methods: in particular, what is the size of the sentence embeddings of "+STN +Fr +De +NLI" and variants?

---

> ### Public Comment · (anonymous) · 2018-01-04
> **size of embeddings - clarification**
>
> Hi,
>
> Thank you for your questions,
>
> In Tables 2, 4, Appendix 6 & Appendix 7 we use the concatenation of the representations produced by two separately trained multi-task GRUs with the strategy described in Appendix section 9. In Table 5, the sentence representations were produced by a single multi-task GRU instead of adopting the concatenation strategy from Appendix section 9, since we wanted to isolate the impact of adding new tasks on our sentence representations. Table 3 evaluates only word embeddings, and not sentence representations, using a single multi-task model. The size of the sentence embeddings of "+STN +Fr +De +NLI" is 3,000; more generally, our models that do not have a +L in Tables 2, 4, Appendix 6 & Appendix 7 are of size 3,000 and the ones that do have a +L or +2L are of size 4,096. We did not add the representation dimensions in Table 2 since we ran out of space (horizontally), but for comparison, the dimensions of Infersent and Skipthoughts are 4096, 4800 respectively.

---

> > ### Author Response · Authors · 2018-01-04
> > **Clarification regarding the comment above**
> >
> > Just wanted to point out that the above comment was made by the authors but doesn't show up as such since we commented from an alternate openreview ID that wasn't associated with this paper.

---

### Author Response · Authors · 2018-01-05
**Some revisions to the paper**

We would like to thank our reviewers and the community in general for their constructive comments and feedback. We have made a few revisions to our paper addressing a few of the concerns raised:

- To quantify transfer performance with a single number, we use the mean difference of our model from Infersent (AllNLI) in Table 2 across all 10 tasks, this number clear illustrates the benefits of adding more tasks on the quality of the learned representations.
- We've added three more rows in Table 3 corresponding to the glove840B, charagram and attract-repel embeddings.
- We've added two more rows in Appendix Table 7 to compare sentence embeddings of different sizes in a non-parametric way.
- We've added a new row to Table 2 (+STN) that corresponds to our implementation of skipthoughts that predicts only the next sentence given the current.
- We've added two more competitive supervised baseline approaches (Neural Semantic Encoder & BLSTM-2DCNN) trained from scratch in Table 2. This is to give readers an idea of the gaps that still exist between transfer approaches and those that learn from scratch.

---

### Decision · Program_Chairs · 2018-01-29
**ICLR 2018 Conference Acceptance Decision**

**Decision:**

Accept (Poster)

**Comment:**

This paper presents a very cool setup for multi task learning for learning fixed length representations for sentences.  Although the authors accept the fact that fixed length representations may not be suitable for complex, long pieces of text (often, sentences), such representations may be useful for several tasks.  They use a significantly large scale setup with six interesting tasks and show that learning generic representations for sentences across tasks is useful than learning in isolation.  Two out of three reviewers presented extensive critique of the paper and there's thorough back and forth between the reviewers and the authors.  The committee believes that this paper will add positive value to the conference.